

# Conserving relics from ancient underground worlds: assessing the influence of cave and landscape features on obligate iron cave dwellers from the Eastern Amazon

Rodolfo Jaffé[1,2,3], Xavier Prous[4], Allan Calux[4], Markus Gastauer[1], Gilberto Nicacio[1], Robson Zampaulo[4], Pedro W.M. Souza-Filho[1,5], Guilherme Oliveira[1], Iuri V. Brandi[4] and José O. Siqueira[1]

[1] Instituto Tecnológico Vale, Belém, PA, Brazil
[2] Ecology, Universidade de São Paulo, São Paulo, Brazil
[3] Ecology, Universidade Federal do Pará, Belém, Pará, Brazil
[4] Environmental Licensing and Speleology, Vale, Nova Lima, Minas Gerais, Brazil
[5] Geoscience, Universidade Federal do Pará, Belém, Pará, Brazil

Corresponding author
Rodolfo Jaffé, r.jaffe@ib.usp.br, rodolfo.jaffe@itv.org

## ABSTRACT

The degradation of subterranean habitats is believed to represent a serious threat for the conservation of obligate subterranean dwellers (troglobites), many of which are short-range endemics. However, while the factors influencing cave biodiversity remain largely unknown, the influence of the surrounding landscape and patterns of subterranean connectivity of terrestrial troglobitic communities have never been systematically assessed. Using spatial statistics to analyze the most comprehensive speleological database yet available for tropical caves, we first assess the influence of iron cave characteristics and the surrounding landscape on troglobitic communities from the Eastern Amazon. We then determine the spatial pattern of troglobitic community composition, species richness, phylogenetic diversity, and the occurrence of frequent troglobitic species, and finally quantify how different landscape features influence the connectivity between caves. Our results reveal the key importance of habitat amount, guano, water, lithology, geomorphology, and elevation in shaping iron cave troglobitic communities. While mining within 250 m from the caves influenced species composition, increasing agricultural land cover within 50 m from the caves reduced species richness and phylogenetic diversity. Troglobitic species composition, species richness, phylogenetic diversity, and the occurrence of frequent troglobites showed spatial autocorrelation for up to 40 km. Finally, our results suggest that the conservation of cave clusters should be prioritized, as geographic distance was the main factor determining connectivity between troglobitic communities. Overall, our work sheds important light onto one of the most overlooked terrestrial ecosystems, and highlights the need to shift conservation efforts from individual caves to subterranean habitats as a whole.

## INTRODUCTION

Caves harbor a unique biodiversity, often comprising obligate subterranean dwellers which must complete their entire life cycle underground (also known as troglobites) (*Pipan & Culver, 2013*). The unparalleled nature of subterranean environments has facilitated the evolution of many endemic troglobites, some of which are considered relics of ancient worlds because their closest relatives have long disappeared from surface environments (*Culver & Pipan, 2009*; *Juan et al., 2010*). Many of these troglobitic species are considered *short-range endemics* (*Harvey, 2002*) because they have only been found in a few caves. As dispersal is assumed to be restricted in these organisms, the degradation of subterranean habitats is believed to represent a threat for the conservation of such short-range endemics. Rare troglobites have therefore been the primary targets of cave conservation efforts worldwide (*Harvey et al., 2011*; *Wynne & Voyles, 2013*; *Culver & Pipan, 2014*; *Ferreira, Oliveira & Silva, 2015*).

Environmental protection agencies of many countries prioritize the conservation of threatened troglobites (*Guzik et al., 2011*; *Harvey et al., 2011*; *Auler & Piló, 2015*), and require extensive speleological surveys prior to the implementation of mining and infrastructure projects. Brazil has one of the most stringent cave protection regimes in the world, which requires government agencies and consulting companies to categorize caves into one of four relevance categories (maximum, high, mid, or low), based on a complex set of biological, geological, and cultural attributes (*Auler & Piló, 2015*). Such categorization is later checked by the environmental protection agencies. Caves containing rare endemic troglobitic species, for instance, are always defined as *maximum relevance* caves, which must be protected along with a buffer area of 250 m (*Jaffé et al., 2016*). *High relevance* caves, on the other hand, can be impacted if appropriate compensation offsets are provided (i.e., preserving two similar caves). Since this protection regime is strictly enforced, the protection of maximum and high relevance caves essentially directs large development projects such as mining operations (Fig. 1).

The factors influencing cave biodiversity remain largely unknown (*Pipan & Culver, 2013*; *Culver & Pipan, 2014*), and the impact of the surrounding landscape on terrestrial troglobitic communities has never been systematically assessed (*Hutchison et al., 2016*; *Pellegrini et al., 2016*). However, a recent study found that habitat (karst) amount and landscape rugosity can predict the presence of major faunal groups of cave obligate species (*Christman et al., 2016*). Previous studies have reported spatial autocorrelation in the number and occurrence frequency of troglobites (*Christman et al., 2005*; *Jaffé et al., 2016*), which suggests some level of subterranean dispersal through porous rocks or micro-cavities (*Auler et al., 2014*). However, the factors influencing subterranean connectivity remain unexplored. Knowledge gaps are even larger in the tropics, where most troglobites remain unidentified to the species level, their distribution ranges have been established based on limited sampling, and their dispersal mechanisms are yet to be determined (*Trajano & Bichuette, 2010*; *Auler & Piló, 2015*; *Jaffé et al., 2016*). For instance, of the 150 troglomorphic species known to be associated to Brazilian iron caves, only 10 have been formally described,

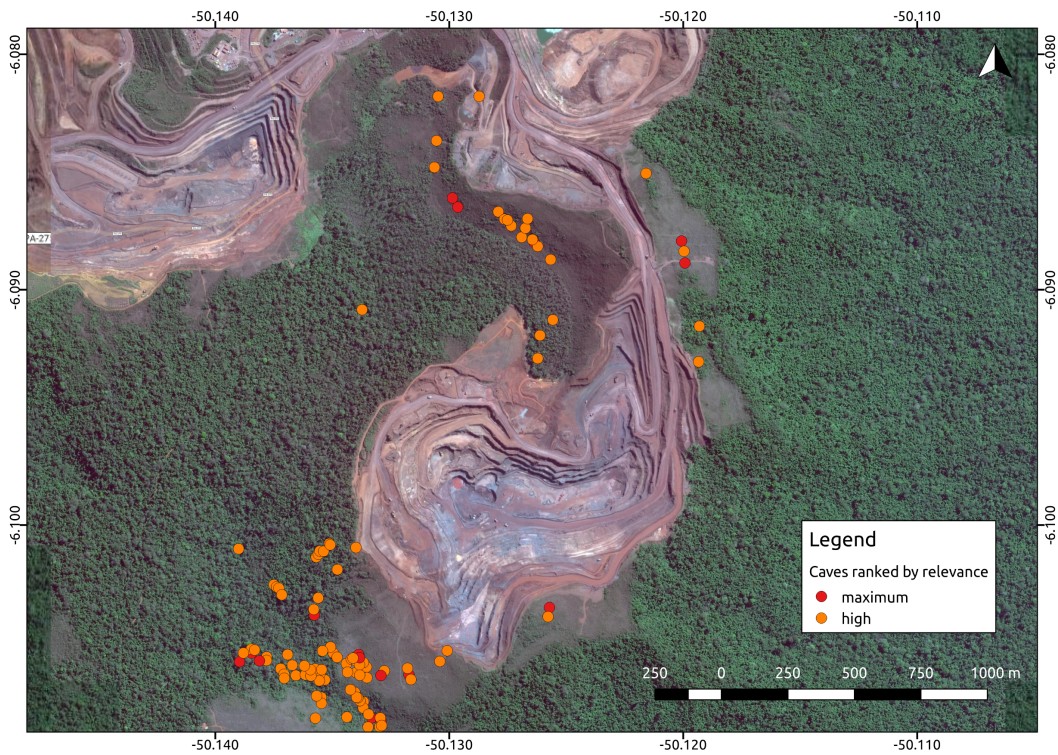

**Figure 1** **Iron ore mine (N5, Serra Norte, Carajás, Brazil) showing the location of caves colored by their relevance classification.** The photo shows how mine planning is affected by the occurrence of maximum relevance caves, which must be protected along with a 250 m radius. Cave data was retrieved from *Jaffé et al. (2016)*, while the image was provided by Google Imagery 2017. Coordinates are shown in decimal degrees.

and include spiders, isopods, springtails, beetles, true bugs and millipedes (*Ferreira, Oliveira & Silva, 2015*).

Here we aim to fill some of these knowledge gaps taking advantage of the most comprehensive speleological database yet available for tropical caves. The Carajás mineral province (South-Eastern Amazon, Brazil) contains one of the world's largest deposits of high-grade iron ore (*Poveromo, 1999*) and some of the largest iron ore mining projects are operating in the region. The environmental licensing of these mines required extensive speleological surveys, undertaken by different companies during the past decade. These surveys generated a wealth of data on cave biodiversity and geological characteristics of iron caves, which not only contain higher richness of troglomorphic species than caves of other lithologies (*Silva, Martins & Ferreira, 2011*), but are also amongst the most threatened subterranean ecosystems. The present study assesses how cave and landscape features influence community composition, species richness, phylogenetic diversity, occurrence and connectivity patterns using previously unpublished and throughly curated lists of troglobitic species for 473 iron caves. We hope the generated knowledge will help guide sound conservation efforts and achieve the best compromise between mining and the protection of cave biodiversity.

Specifically, our study aims to: (1) assess the influence of cave characteristics and the surrounding landscape on troglobitic species composition, species richness, phylogenetic diversity, and the occurrence of frequent troglobitic species; (2) determine the spatial pattern of troglobitic community composition, species richness, phylogenetic diversity, and the occurrence of frequent troglobitic species; and (3) quantify how different landscape features influence the connectivity between caves.

## MATERIALS & METHODS

*Jaffé et al. (2016)* used data from 844 iron caves to assess how the current relevance classification scheme ranks caves with different biodiversity indicators. Additionally, they modelled total species richness (considering all taxa found inside caves), and the presence/absence of rare troglobites, troglobites and bat populations. In the present work, we retrieved previously unpublished data from a curated database containing comprehensive lists of all troglobitic taxa occurring in 473 iron caves of Carajás, State of Pará, Brazil. We then use this dataset to assess the influence of cave characteristics and the surrounding landscape on troglobitic communities, model troglobitic community composition, troglobitic species richness, troglobitic phylogenetic diversity, and the occurrence of frequent troglobitic species, and finally quantify how different landscape features influence the connectivity between caves.

### Dataset

Initially, species inventories were obtained from speleology reports prepared by independent consulting companies. All but one report (N5M2) have already been submitted to the corresponding government agencies (the Brazilian Environmental Protection Agency—IBAMA, and the Pará State's Environment Agency—SEMAS-PA), and are available as Supporting Information in *Jaffé et al. (2016)*. Consulting companies employed similar sampling methods and evaluated the same set of cave attributes (as specified in the Brazilian legislation for the protection of caves: Federal Decree 6640/2008 and Normative Instruction MMA 02/2017). Cave terrestrial fauna, for instance, was surveyed through the full extension of each cave, once during the dry season and once during the wet season. Species lists were later validated by specialized taxonomists, who compared collected specimens across caves and identified them to the finest possible taxonomic level. Still, many specimens were left to the morpho-species level because they represent new undescribed taxa (*Trajano & Bichuette, 2010*). Taxonomists classified species as troglobites if they exhibited troglomorphic traits (*Pipan & Culver, 2013*) absent in phylogenetically related taxa occurring in above-ground habitats. Non-troglobitic species were not compared across caves so they were excluded from our dataset. Additionally, personnel from the Department of Environmental Licensing and Speleology from Vale (a mining company), traveled to caves containing rare troglobites to confirm occurrences and collect specimens in surrounding caves, aiming at validating their occurrence range. We then gathered all available information on the cave's characteristics from different speleology reports and Vale's speleology database. These included cave coordinates, altitude, horizontal projection (length), slope, area, volume, presence of percolating water

and water reservoirs, presence of plant material, presence of plant detritus, presence of roots, presence of guano, presence of other feces, presence of regurgitation balls, presence of carcases, and presence of resident bat populations. We also assessed the external environment by calculating additional cave and landscape metrics at four different spatial scales (50, 100, 250, and 500 m radii from the cave's centroid). These metrics included subterranean area, cave density, cave declivity, lithology, distance to nearest creek, geomorphology, and land cover (See Table S1). Water bodies and geomorphology maps were obtained from the Brazilian Institute of Geography and Statistics (IBGE: ftp://geoftp.ibge.gov.br/cartas_e_mapas/bases_cartograficas_continuas/bcim/), *Souza-Filho et al. (2016)* provided a high resolution 2013 land cover map, and all other metrics were calculated using data from Vale's speleology database. We could not retrieve all metrics in all caves, so some were excluded from certain analyses. The full datasets, including the geographic coordinates of all caves, can be found along with all R scripts as (Dataset S1). All statistical analyses were implemented in R (*R Core Team, 2015*).

## Modeling troglobitic species composition, species richness and phylogenetic diversity

To avoid possible biases arising from unequal sampling efforts across caves, we only considered presence/absence data, omitting abundance information. We first created a community matrix, containing information on the presence or absence of a given species in each cave. We then used the vegan R package (*Oksanen et al., 2016*) to run a Principal Coordinates Analysis (PCoA), employing Bray–Curtis dissimilarity distances. The first two axis of this PCoA (MDS1 and MDS2) were considered proxies of troglobitic community composition, and thus used as response variables in subsequent composition models (Fig. S1). While troglobitic species richness was estimated as the total number of species present in each cave, rarity-weighted richness was also calculated to account for differences in species representation across caves (*Albuquerque & Beier, 2015*). Additionally, phylogenetic diversity was used as a proxy of functional diversity (*Flynn et al., 2011*; *Faith, 2015*). We constructed a phylogenetic tree containing all troglobitic families sampled in at least one cave using the Timetree of Life (*Hedges & Kumar, 2009*; *Hedges et al., 2015*). In its actual version, the Timetree of Life contains more than 50,000 species, and all families from our set of troglobitic species are represented. We inserted genera and subsequently species this phylogeny, considering cases with more than two genera per family or more than two species per genus as polytomies. The resulting phylogeny was dated using the *bladj* algorithm from Phylocom-4.2 in combination with mean age estimates of 29 internal nodes retrieved from the databases (*Hedges & Kumar, 2009*; *Hedges et al., 2015*) (Table S2). Nodes without age estimates were placed evenly between two dated nodes (Fig. S2). Phylogenetic diversity was finally calculated for each cave using Phylocom 4.2 (*Webb et al., 2002*), as the sum of the lengths of all branches considered members of the corresponding minimum spanning path.

We assessed the influence of internal (cave characteristics) and external (surrounding landscape) features on our five response variables (species composition axes MDS1 and MDS2, species richness, log-transformed rarity-weighted richness and phylogenetic

diversity). To do so we used the *lme4* package (*Bates et al., 2014*) to fit linear mixed models (for species composition, rarity-weighted richness and phylogenetic diversity) and generalized linear mixed models (GLMM) with Poisson distributed errors (for species richness), always keeping the geological unit (or highland) where the caves were located as a random effect. This allowed us to account for spatial autocorrelation as well as other potential unmeasured site-specific covariates. We first reduced the large number of initial predictor variables (see Table S1) by identifying the relevant scale at which each landscape component best explained our five response variables. This was done by comparing simple models containing the same predictor measured at the four different spatial scales. Model selection was based on the Akaike Information Criterion (AIC), and the best model (with the lowest AIC) was considered to represent the relevant scale for a given predictor. Because lithology and geomorphology contained many different classes, we ran principal component analyses for each group of variables and included the first two synthetic axis in our models. We then fit full models containing predictors at the relevant scale for each response variable, along with the largest possible number of uncorrelated covariates ($r < 0.6$). All models contained more than 20 observations for every predictor variable, so overfitting was not an issue. Likelihood ratio tests (LRT) were employed to compare full models with reduced models where each predictor was removed one by one. Only predictor variables that significantly improved the full model's log-likelihood (at $\alpha < 0.05$) were included in the final best models. These were then validated by plotting residual vs. fitted values, residual vs. predictors, and by looking at the distribution of residuals. We also tested for spatial autocorrelation in the final model's residuals at the minimum spatial scale (see below).

To assess sampling bias effects we tested if caves located in the proximity of mines (which influence mine planning) were sampled more or less intensively than more distant caves (which do not influence mine planning). To do so we modeled total species richness (all taxa recorded inside each cave), troglobitic species richness and presence of rare troglobites (found in three or less caves) in relation to the mining area surrounding caves at different spatial scales (distant caves had mining areas of zero). Different species richness and different probabilities of containing rare troglobites in caves surrounded by larger mining areas would indicate uneven sampling efforts (since more species are likely to have been identified and occurrence areas of rare species expanded with larger sampling efforts). On the contrary, similar richness and rarity patterns across all caves would reveal no systematic sampling bias effects. We ran GLMM with Poisson distributed errors (for species richness) and GLMM with Bernoulli distributed responses (for presence of rare troglobites), keeping the highland where the caves were located as a random effect. Since mining area surrounding the caves was correlated across spatial scales, we only constructed models containing a single predictor (mining area at a given scale).

## Modeling the occurrence of frequent troglobites

To unravel which cave characteristics help predict the occurrence of certain troglobitic species we analyzed a subset of our data containing the most frequent species (occurring in at least 30 caves). We decided not to analyze species occurring in fewer than 30 caves
to avoid overfitting and complete separation problems resulting from small sample sizes. The presence of each one of these frequent species was modeled using generalized linear models with Bernoulli distributed responses (logistic regressions). Presence/absence was thus set as response variable, and all meaningful uncorrelated cave attributes as predictors. As described above, LRT were employed to identify which predictor variables improved the model's log-likelihood.

## Assessing spatial autocorrelation

We assessed spatial autocorrelation in our five response variables (species composition axes MDS1 and MDS2, species richness, rarity-weighted richness and phylogenetic diversity). The package *spdep* (*Bivand & Piras, 2015*) was employed to estimate Moran's I, a standard measure of spatial autocorrelation ranging from −1 (indicating perfect dispersion) to +1 (perfect correlation, with zero indicating a random spatial pattern). As Moran's I is affected by the spatial scale chosen to assign weights to neighbors, we quantified spatial autocorrelation across the full range of spatial scales of our data. We also tested for spatial autocorrelation in the presence of each one of the frequent troglobitic species. To do so we employed the Join Count Test of the *spdep* package (*Bivand, Pebesma & Gómez-Rubio, 2008*) and assessed spatial autocorrelation across the full range of spatial scales of our data. The Single Color Statistic was computed for presence-presence in networks of neighboring caves located within increasing distances, until we reached the maximal extent of our study region.

## Modeling connectivity between caves

In order to assess how different landscape features influence the connectivity between caves, we used a landscape genetics approach, whereby a dissimilarity measure was related to landscape resistance to dispersal (*Jaffé et al., 2015*). Community dissimilarity (Bray–Curtis distance) was used as a proxy for connectivity, assuming that connected caves have more similar troglobitic communities than isolated ones. We then relied on circuit theory (*McRae, 2006*) to estimate landscape resistance to dispersal between caves, considering land cover, elevation, terrain roundedness, and geographic distance.

Because troglobites are obligate subterranean dwellers (*Pipan & Culver, 2013*; *Culver & Pipan, 2014*), we assumed they can only disperse through the shallow subterranean habitat formed by lateritized igneous mafic rocks and banded ironstone formations (also known as Mountain Savanna or Canga), where our caves are found. We therefore used a 2013 land cover classification map (*Souza-Filho et al., 2016*) (see Fig. 2) and created a resistance surface to where we attributed low resistance values (0.1) to Canga pixels and high resistance values (0.9) to all other pixels (all other land cover classes). To test whether lower elevations represented higher resistance to troglobite dispersal (given these organisms occur in the highlands), we used a Digital Elevation Model—DEM (SRTM 1 Arc-second global downloaded from https://earthexplorer.usgs.gov/) to build a resistance surface where high elevation pixels represented lower resistance than lowland areas. Additionally, we used the same DEM to create a terrain ruggedness raster using the Terrain Analysis plugin in QGIS V2.14, and test whether pixels with higher roundedness represent higher resistance
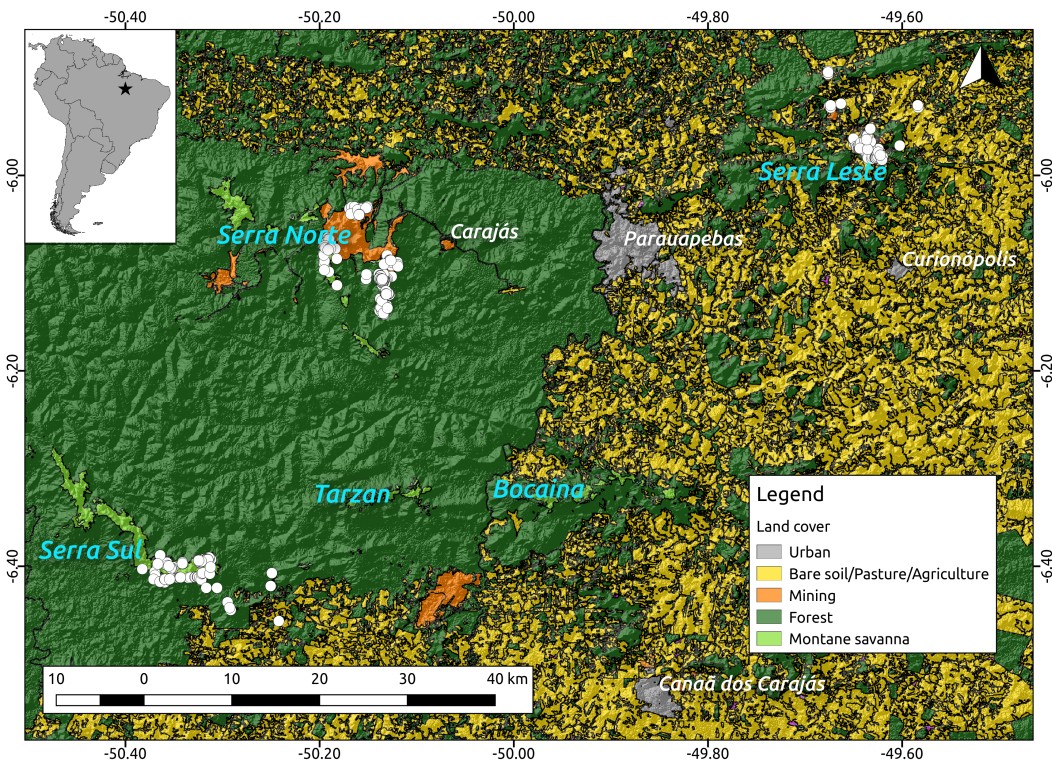

**Figure 2** **Location of the study region (upper left corner) and a detail of the study area showing the spatial distribution of the caves included in our analyses (white dots, $N = 473$) over elevation and a land cover layer.** While the digital elevation raster (SRTM, 1 arc-second) was obtained from USGS Earth Explorer, the land use classification shapefile was obtained from *Souza-Filho et al. (2016)*. Coordinates are shown in decimal degrees.

to troglobitic dispersal. Finally, we created a null-model raster (isolation by geographic distance), where all pixels were coded with identical resistance values (0.5). All rasters were cropped to the extent of the cave locations plus a buffer area of 5 km to minimize border effects (*Jaffé et al., 2015*).

Using the program Circuitscape V4.0 (*McRae, 2006*) we then calculated pairwise resistance distances between all caves, employing all the resistance surfaces described above (land cover, elevation, ruggedness and isolation by geographic distance). Due to Circuitscape's computing limitations we replaced zero values in all rasters with 0.001, and decreased the resolution of all rasters to achieve reasonable computing times. Finally, we regressed Bray–Curtis dissimilarity distance against resistance distances using Maximum Likelihood Population Effects (MLPE) (*Clarke, Rothery & Raybould, 2002*) to account for the non-independence of pairwise distances (*Jaffé et al., 2015*). Code implementing the MLPE correlation structure within the R package *nlme* (*Pinheiro et al., 2014*) is provided at (https://github.com/nspope/corMLPE). Because all resistance distances were highly correlated, we only ran simple MLPE models and compared them using the sample size corrected Akaike Information Criterion (AICc).

## RESULTS

The composition of troglobitic communities was influenced by the total subterranean area, cave density, terrain declivity, altitude, lithology, mining and Canga area (Table 1). Specifically, larger mining areas surrounding caves were associated with the occurrence of Paronellidae *sp.4*, whereas caves surrounded by smaller mining areas were usually inhabited by *Charinus carajas* and Pyrgodesmidae *sp.1* (Fig. S1). Similarly, caves surrounded by larger Canga areas were associated with *Charinus carajas*, whereas caves surrounded by smaller areas of Canga usually contained more Systrophiidae *sp.1*. The relevant scale at which mining and Canga influenced community composition differed, being 250 m for the former and 50 m for the later (Fig. 3). Species richness, rarity-weighted richness and phylogenetic diversity were all explained by the distance to the nearest creek, geomorphology, cave area and the presence of guano (Table 1). However, agriculture land cover was found associated to both species richness and phylogenetic diversity, while Canga land cover was an important predictor of rarity-weighted richness. Subterranean area was also found associated to phylogenetic diversity (Table 1). Interestingly, species richness, rarity-weighted richness and phylogenetic diversity increased with increasing distance to the nearest creek, increasing cave area, and the presence of guano. On the other hand, the amount of agricultural landscapes surrounding caves was negatively associated with both species richness and phylogenetic diversity, and in both cases the relevant scale for agriculture land cover was 50 m (Fig. 3). Finally, rarity-weighted richness increased with the amount of Canga land cover surrounding caves and the relevant scale for the effect of Canga land cover was 50 m (Fig. 3). We did not find spatial autocorrelation in any of the model's residuals and no systematic sampling bias effects were detected, given that richness and rarity patterns were not influenced by the cave's proximity to mines (Table S3).

The occurrence of the most frequent species was predicted by cave characteristics, with altitude being the variable determining the presence or absence of most species (Table 2). Presence of guano, cave slope, and presence of water reservoirs were also identified as important predictors of some frequent species, although the direction of these effects varied between species (Table 2).

Species composition, species richness, rarity-weighted richness and phylogenetic diversity were found to be spatially autocorrelated for up to 40 km (Fig. 4). Above this distance spatial autocorrelation disappeared, revealing a random spatial pattern. The occurrence of frequent troglobitic species was also found to be spatially autocorrelated across spatial scales (Fig. 5). Most troglobitic species were found to be restricted to one or a few caves, but a few were found in more than 100 caves (Fig. 5).

Our connectivity analyses revealed that the model containing geographic distance was the best to explain community dissimilarity, while neither land cover, terrain ruggedness or elevation seemed to influence community dissimilarity (Table 3). Specifically, dissimilarity increased with increasing geographic distance separating caves (Fig. 6).

**Table 1** Summary of the best models describing troglobitic species composition, species richness, rarity-weighted richness and phylogenetic diversity.

| Response variable | Model[a] | Predictor variables | Estimate | SE | t/z | p |
|---|---|---|---|---|---|---|
| Species composition (MDS1) | LMM | Subterranean area (500 m) | 0.13 | 0.04 | 3.12 | 0.002 |
| | | Declivity 0–10° (500 m) | 0.09 | 0.03 | 2.98 | 0.003 |
| | | Mining area (250 m) | −0.08 | 0.03 | −3.01 | 0.003 |
| Species composition (MDS2) | LMM | Declivity 30–40° (500 m) | −0.06 | 0.02 | −2.74 | 0.006 |
| | | Cave density (50 m) | −0.08 | 0.02 | −3.23 | 0.001 |
| | | Lithology axis 2 (50 m) | −0.10 | 0.04 | −2.51 | 0.012 |
| | | Canga area (50 m) | 0.07 | 0.03 | 2.44 | 0.015 |
| | | Altitude | −0.06 | 0.03 | −1.82 | 0.070 |
| Species richness | GLMM | Distance to nearest creek | 0.13 | 0.04 | 3.04 | 0.002 |
| | | Geomorphology axis 2 (50 m) | 0.20 | 0.05 | 3.85 | <0.001 |
| | | Agriculture area (50 m) | −0.15 | 0.04 | −4.29 | <0.001 |
| | | Cave area | 0.30 | 0.03 | 10.11 | <0.001 |
| | | Presence of guano | 0.35 | 0.07 | 5.33 | <0.001 |
| Rarity-weighted richness (log-transformed) | LMM | Distance to nearest creek | 0.28 | 0.1 | 2.87 | 0 |
| | | Geomorphology axis 2 (50 m) | 0.27 | 0.11 | 2.56 | 0.01 |
| | | Canga area (50 m) | 0.25 | 0.08 | 3.34 | <0.001 |
| | | Cave area | 0.5 | 0.07 | 7.55 | <0.001 |
| | | Presence of guano | 0.54 | 0.14 | 3.89 | <0.001 |
| Phylogenetic diversity | LMM | Distance to nearest creek | 176.25 | 64.69 | 2.72 | 0.007 |
| | | Subterranean area (50 m) | 125.46 | 49.04 | 2.56 | 0.011 |
| | | Geomorphology axis 2 (50 m) | 180.44 | 71.08 | 2.54 | 0.012 |
| | | Agriculture area (50 m) | −203.45 | 51.50 | −3.95 | <0.001 |
| | | Cave area | 551.18 | 53.99 | 10.21 | <0.001 |
| | | Presence of guano | 390.94 | 93.70 | 4.17 | <0.001 |

**Notes.**

Response variables are shown followed by the type of model employed (Model), the predictor variables included in the best models (selected through likelihood ratio tests with $\alpha$ <0.05), estimates, standard errors (SE), t/z-values and p-values.

[a] Linear mixed models (LMM) or Generalized Linear Mixed Models (GLMM). All models accounted for spatial autocorrelation and unaccounted variation between study sites by keeping the highland were the caves were located as a random effect. Eight out of the 473 caves were excluded from these analyses because they contained missing data in at least one field.

# DISCUSSION

Relying on spatial statistics to analyze the most comprehensive speleological database yet available for tropical caves, our study reveals the factors underpinning troglobitic community composition, species richness, and phylogenetic diversity. Additionally, we assess patterns of spatial distribution and provide the first insights into the factors influencing the connectivity of troglobitic communities from the Amazon's main source of iron ore.

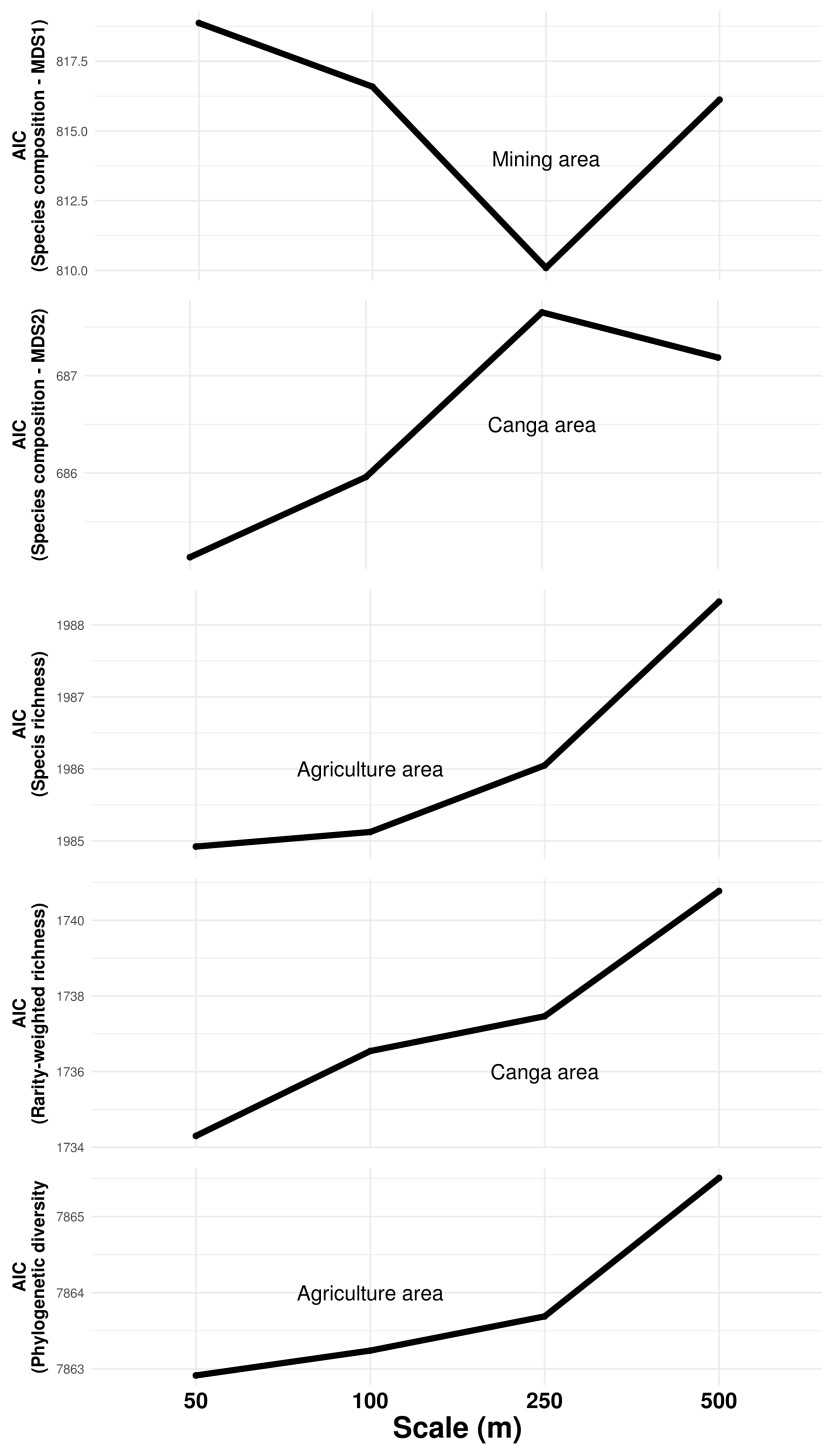

**Figure 3 Influence of land cover on troglobitic species composition, species richness, rarity-weighted richness and phylogenetic diversity across spatial scales.** The *Y* axis show the Akaike Information Criteria (AIC) of simple mixed models containing each land cover predictor at four different spatial scales. Lowest AIC values indicate the scale at which each predictor best explained response variables.

**Table 2 Parameter estimates for the best models explaining the presence of the most frequent troglobitic species (occurring in at least 30 caves).**

| Species | Altitude | Presence of guano | Cave slope | Presence of water reservoirs | Presence of other feces | Cave area |
|---|---|---|---|---|---|---|
| aff Xyccarph sp1 | −0.003 | −0.947 | | | | |
| Carajas paraua | | | 0.136 | 0.642 | | |
| Charinus carajas | −0.002 | | 0.107 | | 0.331 | |
| Charinus orientalis | −0.004 | 1.230 | −0.164 | | | |
| Circoniscus buckupi | 0.003 | | | −1.192 | 1.600 | |
| Cyphoderidae sp1 | | 1.106 | | | | |
| Cyphoderidae sp2 | | 1.753 | | | | 0.001 |
| Entomobryidae sp4 | 0.007 | | | | | |
| Entomobryomorpha sp | −0.004 | | 0.156 | 1.172 | −1.316 | |
| Isotomidae sp2 | −0.003 | | 0.102 | 1.035 | | |
| Matta sp1 | 0.009 | | 0.104 | | | |
| Paronellidae sp4 | 0.006 | 0.708 | | | | 0.001 |
| Pyrgodesmidae sp1 | −0.004 | −0.900 | 0.159 | 1.341 | −0.833 | |
| Systrophiidae sp1 | −0.010 | 1.705 | | −1.672 | | |
| **Total number of models** | **11** | **7** | **7** | **6** | **4** | **2** |

Species inventories were initially obtained from independent consulting companies, who employed similar sampling methods. These were later validated by specialized taxonomists, and occurrence ranges of rare troglobites were confirmed through further sampling. Since we did not detect any systematic sampling bias effects (Table S3), we are confident that our results reflect the ecology of iron cave troglobites. On the other hand, temporal mismatches between the maps employed to assess landscape features and the timing of speleological surveys could have influenced our analyses. Speleological surveys performed by consulting companies took place between 2005 and 2011 (Serra Norte), between 2010 and 2011 (Serra Leste), and between 2010 and 2011 (Serra Sul; see details in *Jaffé et al., 2016*). While mining activities in Serra Norte began in the 1980s, long before the speleological surveys took place, Serra Leste had also been exposed to cattle ranching before the speleological surveys were conducted there. Serra Sul, on the other hand, was still completely preserved in 2013 as mining activities had not yet begun. Since we calculated landscape metrics using a 2013 land cover map, we believe time-lag effects had a minor impact on our results.

Previous studies have shown that cave size is a key predictor of subterranean biodiversity, because larger caves not only have higher colonization rates, but can also host larger and more diverse communities (*Brunet & Medellín, 2001*; *Silva, Martins & Ferreira, 2011*; *Simões, Souza-Silva & Ferreira, 2015*). In turn, more diverse communities have been found to contain more troglobitic species (*Culver et al., 2004*; *Christman et al., 2005*), and a recent study on iron caves found a strong correlation between total species richness and the richness of troglobites (*Jaffé et al., 2016*). Our results match these previous findings, as habitat amount (assessed through cave area, subterranean area, cave density, and Canga area) was found associated to all response variables (species composition, species richness,

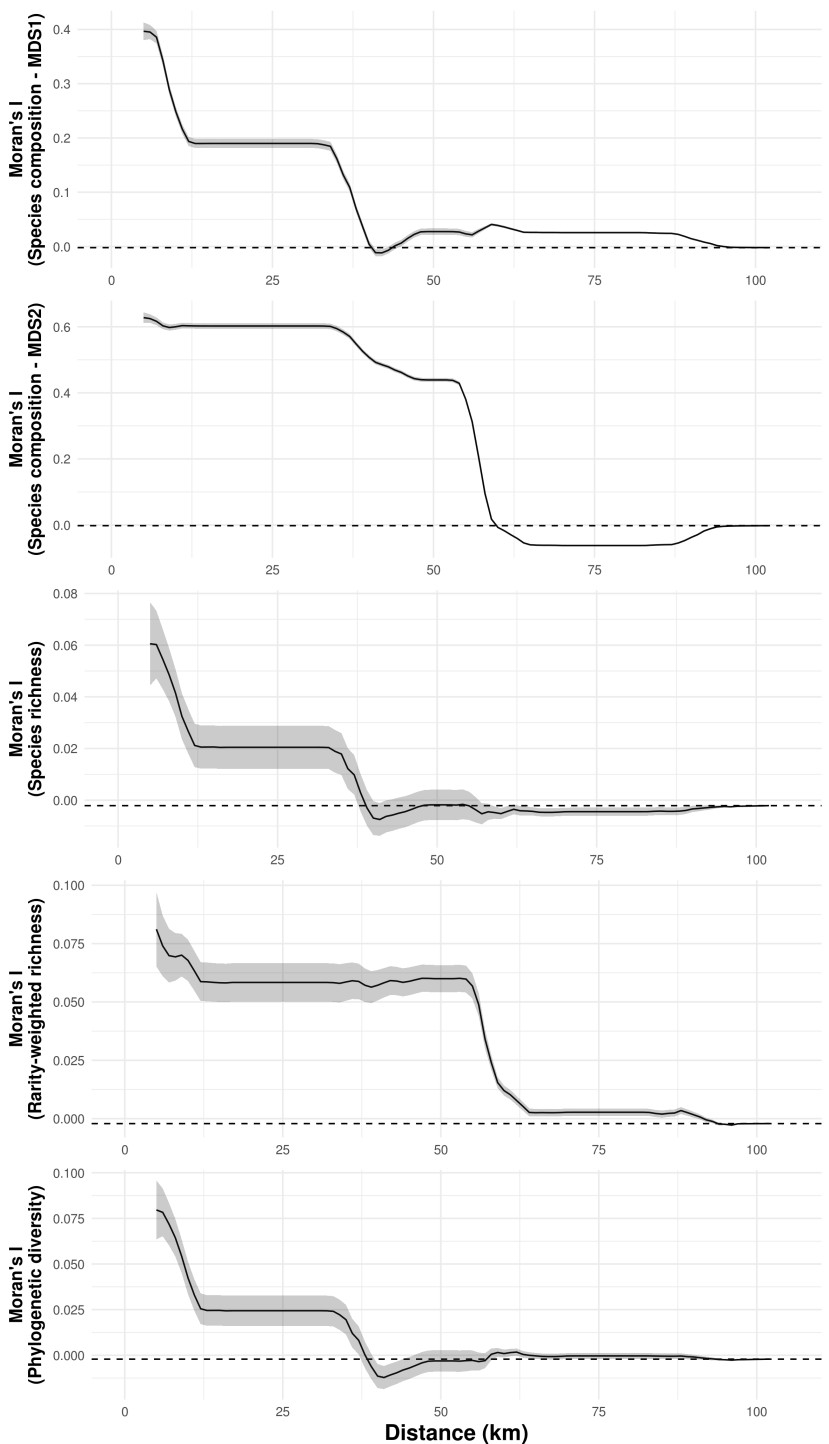

**Figure 4** **Spatial autocorrelation of troglobitic species composition, species richness, rarity-weighted richness and phylogenetic diversity across different spatial scales.** While the solid lines show the value of Moran's I estimates, the gray area depict 95% confidence intervals. The dashed lines represent expected values under a null model of no spatial autocorrelation.

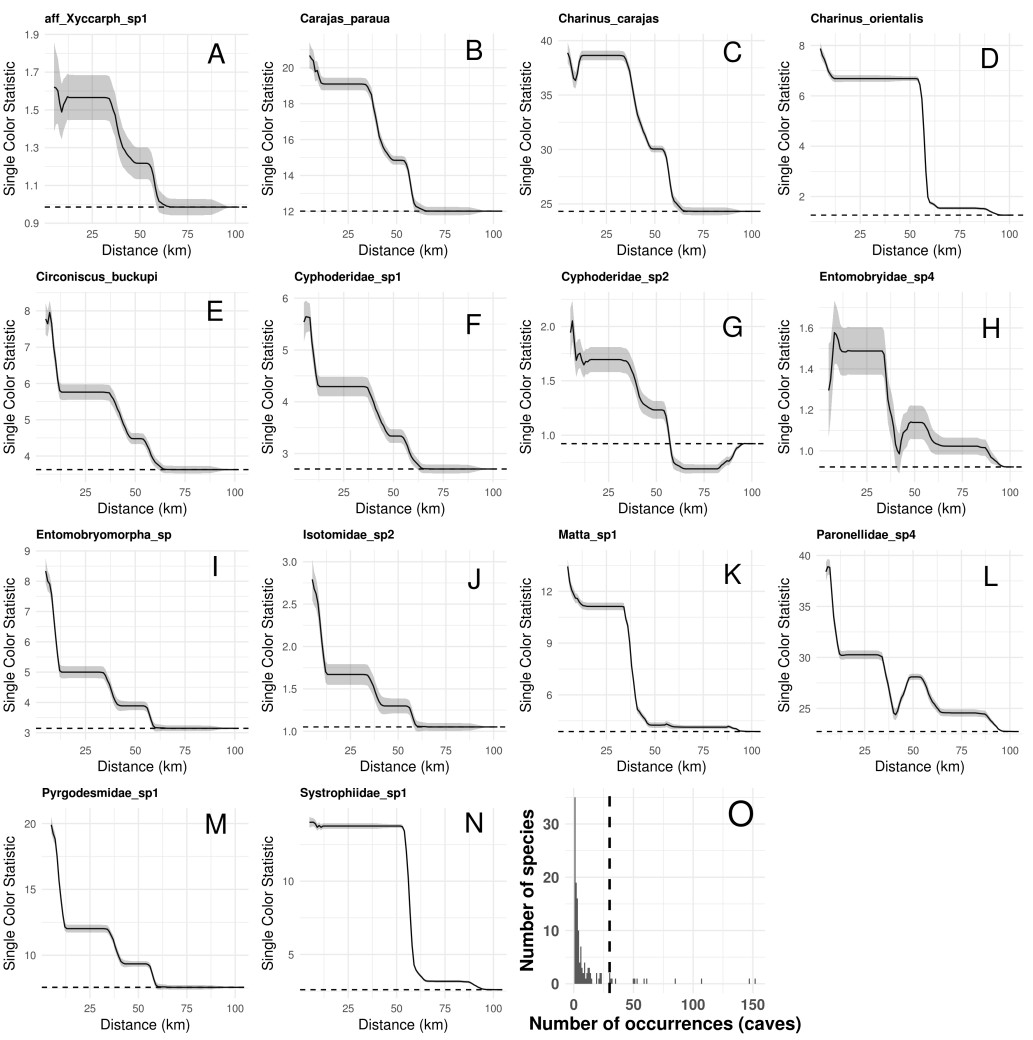

**Figure 5  Spatial autocorrelation in the occurrence of the most frequent troglobitic species (occurring in 30 or more caves) across different spatial scales (A–N), and frequency distribution of occurrences by species (O).** Solid lines in A–N show the value of Single Color Statistic estimates, gray area depict 95% confidence intervals, and dashed lines represent expected values under a null model of no spatial autocorrelation. The dashed vertical line in O shows the threshold value of species occurring in at least 30 caves.

rarity-weighted richness, phylogenetic diversity, and the occurrence of some frequent species). Additionally, our results show that proximal habitat amount (Canga within 50 m of the caves) is an important predictor of troglobitic community composition and the presence of rare species (Fig. 3). Interestingly, our study is the first one to report an association between the amount of subterranean habitat and phylogenetic diversity, our proxy for functional diversity. This finding suggests that more diverse communities are also more complex and possibly more resilient ones, given the higher functional diversity they harbor (*Lean & Maclaurin, 2016*).

Our data also supports the idea that a higher availability of trophic resources facilitates colonization of the cave's interior (*Poulson & White, 1969*; *Culver & Pipan, 2009*;
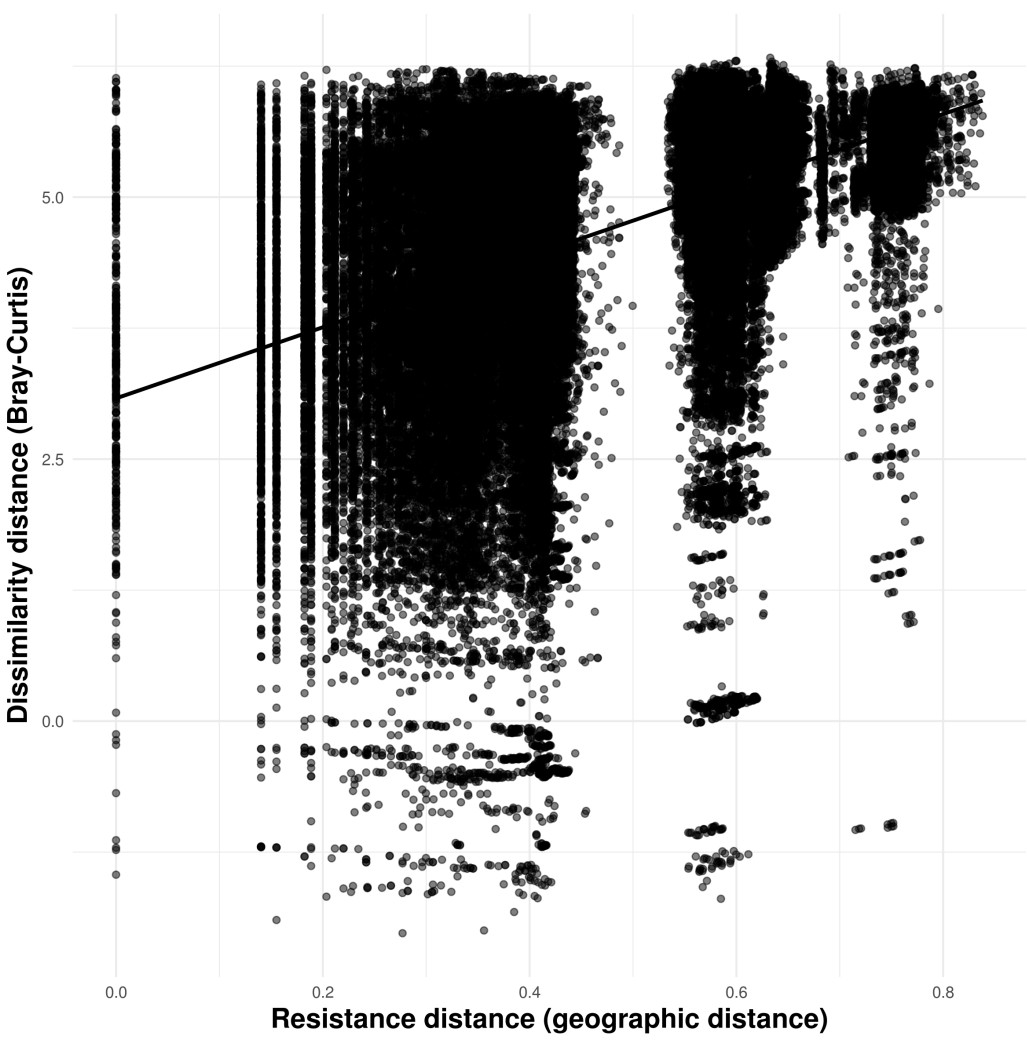

**Figure 6** **Relationship between troglobitic community dissimilarity (Bray–Curtis distance) and geographic distance resistance distance.** Dissimilarity distance is de-correlated for the MLPE correlation structure.

**Table 3** **Comparison of connectivity models.**

| Predictors variable | Log-likelihood | AICc | ΔAICc | Weight |
|---|---|---|---|---|
| Geographic distance | 41,538.18 | −83,068.35 | 0.00 | 1.00 |
| Land cover | 41,392.50 | −82,777.00 | 291.35 | 0.00 |
| Rougedness | 41,352.66 | −82,697.32 | 371.04 | 0.00 |
| Elevation | 40,989.58 | −81,971.17 | 1,097.18 | 0.00 |

*White & Culver, 2012*; *Ferreira, Oliveira & Silva, 2015*), as species richness, rarity-weighted richness and phylogenetic diversity were higher in caves containing guano. For instance, the cave's deep interior has been compared to a desert since it is largely deprived of trophic resources (*White & Culver, 2012*; *Pipan & Culver, 2013*). Troglobites thus rely on

external material that is washed into the cave or brought in by mobile species (*Poulson & White, 1969*; *Taylor, Krejca & Denight, 2005*; *White & Culver, 2012*; *Wynne & Voyles, 2013*). Although our findings match those of a recent analysis of 844 iron caves from the Carajás region, which also found higher species richness in caves containing guano (*Jaffé et al., 2016*), they reveal that this trophic resource not only supports higher species richness but also a higher functional diversity (assessed via phylogenetic diversity). Additionally, our occurrence models reveal that the presence or absence of guano and other feces determines the occurrence of certain frequent species (Table 2), as found for other troglobites (*Ferreira & Martins, 1999*; *White & Culver, 2012*).

We also identified an effect of cave declivity and slope on species composition, which suggests declivity influences the amount of resources that are carried or washed into caves. Interestingly, species richness, rarity-weighted richness and phylogenetic diversity increased with increasing distances to creeks, indicating a negative effect of water on cave diversity. Supporting these findings, a study of 55 limestone caves from the Brazilian Savannah found that the presence of water bodies significantly influences species composition (*Simões, Souza-Silva & Ferreira, 2015*). Similarly, *Jaffé et al. (2016)* found a lower total species richness in caves containing water reservoirs. Although we did not find a general effect of water reservoirs on troglobitic communities, they were found to determine the occurrence of some frequent species (Table 2). Water thus seems to be an important driver of troglobitic biodiversity.

Whereas lithology influenced species composition, geomorphology affected species richness, rarity-weighted richness and phylogenetic diversity. Specifically, the lithology effect was mainly driven by the amount of basic rocks (Fig. S3), while homogeneous or sharp differential geomorphology was the strongest correlate of species richness, rarity-weighted richness and phylogenetic diversity (Fig. S4). Reinforcing our results, a recent study of ten limestone caves found that the amount of limestone outcrops surrounding the caves influence invertebrate community composition (*Pellegrini et al., 2016*). Likewise, a study of 33 caves from Central Italy revealed that cave morphology and microclimate strongly affected the composition of non-strict cave dwelling organisms (*Lunghi, Manenti & Ficetola, 2014*). Finally, a recent landscape genetic study of secret cave cricket populations occurring in 42 limestone caves of central Texas, found a strong influence of karst topography on cricket gene flow (*Hutchison et al., 2016*). Along with our findings, this accumulated evidence highlights the role of lithology and geomorphology in shaping troglobitic communities.

Elevation was another factor found associated with the species composition of troglobitic communities, with species like Systrophiidae *sp.1* occurring at higher elevations and species like *Charinus carajas* occurring at lower ones (Table 1, Fig. S1). Interestingly, altitude was found to be the main variable determining the presence of frequent troglobites, indicating that these species exhibit narrow elevation preferences (elevation ranged from 224 to 842 masl). This result, along with the fact that frequent species responded differently to cave characteristics (Table 2), indicates a high level of specialization, as described for many troglobites (*Culver & Pipan, 2009*; *Pipan & Culver, 2013*).

To our knowledge, ours is the first study revealing an effect of anthropogenic land use on terrestrial troglobitic communities (*Gunn, Hardwick & Wood, 2000*; *Wood, Gunn & Perkins, 2002*; *Moraes, Landis & Molander, 2002*). While mining area within 250 m from the caves influenced species composition, agriculture land cover at the smallest measured scale (50 m) had a significant impact on species richness and phylogenetic diversity. Larger mining areas surrounding caves were associated with the occurrence of Paronellidae *sp.4*, whereas caves containing smaller or no mining areas usually contained *Charinus carajas* and Pyrgodesmidae *sp.1* (Fig. S1). These results suggest that Paronellidae *sp.4* is more resilient to mining-led landscape changes than *Charinus carajas* or Pyrgodesmidae *sp.1*, which seem more susceptible (these three species occur in 147, 152, and 85 caves respectively, so results are not biased by small sample sizes). However, agriculture but not mining land cover was found associated to species richness and phylogenetic diversity (Table 1). This result was unexpected, given the huge impact of mining on Canga subterranean habitats (Fig. 1), and suggests a role of agricultural practices in the observed decay of species richness and functional diversity. Indeed, herbicides, fungicides, insecticides, fertilizers and mineral salt are widely used in the region, and farmers frequently burn pasturelands (*Perz, 2003*). As invertebrates have been found among the most affected group by the pesticide doses employed (*Schiesari et al., 2013*), our findings suggest a role of these compounds in the observed reduction in species richness. Additionally, fire may also influence shallow subterranean environments, and deforestation is likely to reduce the amount of organic material reaching the cave's interior (*Beynen & Townsend, 2005*). Finally, land use changes impacting bat populations (i.e., reducing available trophic or roosting resources) are also likely to affect troglobitic communities by depriving them of guano (*Muylaert, Stevens & Ribeiro, 2016*).

Previous studies have reported spatial autocorrelation in the number of troglobitic species, the number of non-endemics, the number and occurrence frequency of single-cave endemics, the total number of terrestrial species, the presence of troglobites, and the presence of rare troglobites (*Christman et al., 2005*; *Jaffé et al., 2016*), which suggests that troglobitic communities are able to influence the troglobitic composition of neighboring caves. Our results match these findings, as troglobitic species composition, species richness, rarity-weighted richness, phylogenetic diversity, and the occurrence of frequent troglobites showed spatial autocorrelation across a range of spatial scales. For instance, the Canga formations where our study caves are found, are constituted by highly porous rocks that form many micro-cavities and cracks (*Ferreira, 2005*; *Silva, Martins & Ferreira, 2011*; *Auler et al., 2014*). These represent potential subterranean habitats that could serve as dispersal corridors for some troglobitic species (*Jaffé et al., 2016*), or may actually constitute the primary subterranean habitat of these organisms, with caves being convenient sampling sites (*Culver & Pipan, 2014*). Although no study had yet explicitly evaluated connectivity between terrestrial troglobitic communities, there is limited evidence for non-obligate subterranean dwellers (*Pipan & Culver, 2007*; *Carlini et al., 2009*; *Hutchison et al., 2016*). Here we quantify the influence of landscape resistance on the similarity of terrestrial troglobitic communities, and found that geographic distance is the main factor determining community dissimilarity. Importantly, neither land cover, terrain ruggedness or elevation

were found to influence community dissimilarity, indicating that anthropic land uses, rough terrain or elevation gradients may not necessarily represent barriers to subterranean cave connectivity (*Christman et al., 2005*).

Finally, our study highlights the uniqueness of troglobites, as most troglobitic species were found to be restricted to one or a few caves (Fig. 5). These rare species, restricted to a few caves (35 species occurred in a single cave), represent the most threatened *short-range endemics*, so they should be conservation priorities. Further actions are nevertheless needed to increase sampling efforts of *single-cave endemics*, confirm occurrence areas, and validate taxonomic identification. Molecular DNA barcoding tools could contribute increase the accuracy of taxonomic classification and achieve a fast cross-validation of species occurrences across caves (*Juan et al., 2010*).

## CONCLUSIONS

Our results have important implications for the protection of cave biodiversity. First, our findings could guide speleological surveys focus on assessing the most relevant cave characteristics driving troglobitic communities (habitat amount, guano, water, lithology, geomorphology, and elevation). Second, our results highlight the need to regulate agriculture in the vicinity (50 m) of iron caves, as agricultural landscapes were found to have a profound impact on troglobitic biodiversity. Third, our work suggests that the conservation of cave clusters should be prioritized, as geographic distance was the main factor determining connectivity between troglobitic communities. Fourth, we argue that conservation efforts should prioritize species occurring in one or a few caves, and underline the need for further actions to confirm occurrence areas and validate taxonomic identification of *single-cave endemics*. Overall, our work sheds important light onto one of the most overlooked terrestrial ecosystems, and highlights the need to shift conservation efforts from individual caves to subterranean habitats as a whole.

## ACKNOWLEDGEMENTS

We thank Vale's Environmental Licensing and Speleology Department for granting access to its speleology database, and Rafael Melo de Brito for help with GIS analyses.

### Funding

Funding was provided by Instituto Tecnológico Vale and CNPq grants 380535/2017-3 (Gilberto Nicacio) and 307479/2016-1 (Guilherme Oliveira). The funders had no role in study design, data collection and analysis, decision to publish, or preparation of the manuscript.

### Grant Disclosures

The following grant information was disclosed by the authors:
Instituto Tecnológico Vale.
CNPq grants: 380535/2017-3, 307479/2016-1.

## Competing Interests

Guilherme Oliveira is an Academic Editor for PeerJ.

## Author Contributions

- Rodolfo Jaffé, Xavier Prous and Allan Calux conceived and designed the experiments, performed the experiments, analyzed the data, contributed reagents/materials/analysis tools, prepared figures and/or tables, authored or reviewed drafts of the paper, approved the final draft.
- Markus Gastauer and Gilberto Nicacio performed the experiments, analyzed the data, contributed reagents/materials/analysis tools, prepared figures and/or tables, authored or reviewed drafts of the paper, approved the final draft.
- Robson Zampaulo performed the experiments, contributed reagents/materials/analysis tools, authored or reviewed drafts of the paper, approved the final draft.
- Pedro W.M. Souza-Filho performed the experiments, contributed reagents/materials/-analysis tools, prepared figures and/or tables, authored or reviewed drafts of the paper, approved the final draft.
- Guilherme Oliveira authored or reviewed drafts of the paper, approved the final draft.
- Iuri V. Brandi and José O. Siqueira authored or reviewed drafts of the paper, approved the final draft, provided institutional support.

## Data Availability

Datasets and R scripts are included as Supplemental Information.

## Supplemental Information

Supplemental information for this article can be found online at http://dx.doi.org/10.7717/peerj.4531#supplemental-information.

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
