# Peer review of "Conserving relics from ancient underground worlds: assessing the influence of cave and landscape features on obligate iron cave dwellers from the Eastern Amazon"

_PeerJ, doi:10.7717/peerj.4531_

## Round 0.1 · original submission · Minor Revisions

I had no substantive issues with the paper. Please take care of the minor comments of the reviewers and you should be in good shape with it. Be sure to include a rebuttal letter to me as to how you addressed these minor comments.

·

Basic reporting

I thought the authors did an admirable job of reporting their results and I had few comments (on annotated pdf file)

Experimental design

I found the statistical methods to be both thorough and innovative. My only suggestion of substance was to consider using a weighted richness measure where each occurrence for a species occurring n time is weighted 1/n. This weights species equally.

Validity of the findings

As the authors point out, the findings are novel, especially with respect to spatial structure.

Additional comments

Iron caves are small. I have generally thought that the primary habitat was the MSS (canga?) and that caves were sampling sites. I realize the importance of caves under Brazilian law, but I wonder if you could comment on this in the discussion. It actually ties in quite well with your findings.

·

Basic reporting

This manuscript represents an attempt to document the spatial and disturbance variables that might affect troglobitic communities and their constituent species in the iron caves of eastern Brazil. The manuscript is well-written and mostly unambiguous. I have uploaded a version of the manuscript with a few small suggestions highlighted. The context is clearly explained, and the literature cited seems comprehensive.

Experimental design

I am not fully conversant with the statistical packages used in the study, but I cannot detect any major flaws. The research questions are well defined.

Validity of the findings

This is a novel study centered on the controversial topic of the conservation of trogolobitic communities in the face of anthropogenic land uses changes. The findings appear to be well supported and do not over-reach the data or the analyses.

---

## Round 0.2 · accepted · Accept

The reviewers were experts in the field and their comments were minor. Thank you for responding thoroughly to what they did have to say. I think this will be an important paper in the conservation of caves and care faunas.